# Neuroendocrine Regulation of Energy Metabolism Involving Different Types of Adipose Tissues

**DOI:** 10.3390/ijms20112707

**Published:** 2019-06-01

**Authors:** Qi Zhu, Bradley J. Glazier, Benjamin C. Hinkel, Jingyi Cao, Lin Liu, Chun Liang, Haifei Shi

**Affiliations:** 1Program of Physiology and Neuroscience, Department of Biology, Miami University, Oxford, OH 45056, USA; zhuq2@miamioh.edu (Q.Z.); glazieb2@miamioh.edu (B.J.G.); hinkelbc@miamioh.edu (B.C.H.); caoj2@miamioh.edu (J.C.); 2Program of Bioinformatics, Department of Biology, Miami University, Oxford, OH 45056, USA; liul2@miamioh.edu (L.L.); liangc@miamioh.edu (C.L.)

**Keywords:** white adipose tissue, brown adipose tissue, beige adipose tissue, adipokines, batokines, sympathetic nervous system, innervation, denervation, thermogenesis, lipolysis, fatty acid oxidation, high-fat diet, RNA sequencing

## Abstract

Despite tremendous research efforts to identify regulatory factors that control energy metabolism, the prevalence of obesity has been continuously rising, with nearly 40% of US adults being obese. Interactions between secretory factors from adipose tissues and the nervous system innervating adipose tissues play key roles in maintaining energy metabolism and promoting survival in response to metabolic challenges. It is currently accepted that there are three types of adipose tissues, white (WAT), brown (BAT), and beige (BeAT), all of which play essential roles in maintaining energy homeostasis. WAT mainly stores energy under positive energy balance, while it releases fuels under negative energy balance. Thermogenic BAT and BeAT dissipate energy as heat under cold exposure to maintain body temperature. Adipose tissues require neural and endocrine communication with the brain. A number of WAT adipokines and BAT batokines interact with the neural circuits extending from the brain to cooperatively regulate whole-body lipid metabolism and energy homeostasis. We review neuroanatomical, histological, genetic, and pharmacological studies in neuroendocrine regulation of adipose function, including lipid storage and mobilization of WAT, non-shivering thermogenesis of BAT, and browning of BeAT. Recent whole-tissue imaging and transcriptome analysis of differential gene expression in WAT and BAT yield promising findings to better understand the interaction between secretory factors and neural circuits, which represents a novel opportunity to tackle obesity.

## 1. Introduction

Maintaining energy homeostasis is imperative for health maintenance in any living organism, a process regulated by complicated neural circuits and secretory factors. The central nervous system (CNS) receives and integrates a variety of external stimuli from the environment such as diet and temperature, along with internal neural and chemical signals from peripheral tissues indicating energy status and storage. The CNS subsequently sends outward signals via the autonomic nervous system that directly contacts multiple tissues and organs to closely control metabolism and achieve the metabolic homeostasis (Figure 1).

Adipose tissue is one of the primary sites of the regulation of lipid metabolism, which includes three metabolic processes: Lipogenesis involving cell proliferation and uptake of circulating free fatty acids; lipolysis involving hydrolysis of triglycerides into glycerol and free fatty acids; and fatty acid β-oxidation inside the mitochondria [1]. During negative energy balance when energy expenditure exceeds energy intake, such as during exercise or fasting, stored lipids are mobilized via lipolysis to release fatty acids and glycerol which are used by other tissues and organs [2]. During positive energy balance when energy intake exceeds energy expenditure, such as during feeding with a high-fat diet, extra calories are primarily stored as lipids that accumulate in adipose tissues via increased fatty acid uptake and de novo lipogenesis [3,4,5,6].

Chronic dysregulation of energy balance due to energy intake exceeding energy expenditure results in excess lipid storage in adipose tissues, leading to obesity. Currently, 39.8% of adults and 18.5% of children in United States have a BMI over 30 and are classified as obese [7]. A number of environmental factors in Western society, including calorie-dense diet, sedentary lifestyle, and stress collectively contribute to an imbalance of energy homeostasis and development of obesity [8]. Obesity is closely associated with the development of chronic diseases, such as type 2 diabetes, cardiovascular diseases, non-alcoholic hepatic steatosis, and some types of cancer, and thus it is a major health issue [9,10,11]. Because of the global increases in the incidence of obesity and its associated metabolic diseases, it is of great significance to understand mechanisms underlying the regulation of lipid metabolism in adipose tissues. Consequently, increasing attention has been placed on understanding the neuroendocrine regulation of adipose function as a regulatory mechanism of energy metabolism. While the innervation of adipose tissue was first reported in the 1890s, more recent advancements in imaging, tracing, and RNA sequencing have yielded new insight into the innervation of adipose tissue and its function. This has led to the discovery of a number of adipose secretory factors that interact with neural circuits extending from the brain to play important roles in regulating numerous processes in white, brown and beige adipose tissues (WAT, BAT, and BeAT, respectively), including lipid storage and mobilization of WAT, fatty acid oxidation and non-shivering thermogenesis of BAT, and browning of BeAT, which ultimately affect whole-body energy homeostasis.

Because a better understanding of the regulation of lipid metabolism at the level of adipose tissues may offer an innovative opportunity to tackle obesity, we aim to review advances in the understanding of regulation of adipose function at different types of adipose tissues by summarizing historic and recent studies using histological, imaging, genetic, and pharmacological approaches. In addition, we compare expression of some genes related to neuroendocrine regulation of lipid metabolism between WAT and BAT using recent RNA sequencing data obtained from different types of adipose tissues of either lean male mice fed a standard low-fat diet or obese male mice fed a high-fat diet for four weeks [12] (Table 1). Special emphasis is placed on results from genomic analysis of various types of adipose tissues as this helps to understand differences between WAT and BAT, clarify unsettled questions, and develop novel targets for better regulation of energy metabolism.

## 2. Different Types of Adipose Tissues

### 2.1. Location of Adipose Tssues

It is currently accepted that there are three types of adipose tissues; WAT, BAT, and BeAT. WAT is located in both subcutaneous and visceral regions (Figure 2). Some commonly studied WATs in rodent models are subcutaneous inguinal WAT (IWAT), visceral retroperitoneal WAT (RWAT) positioned behind the kidneys [13], mesenteric and omental WAT (MWAT) located in intestinal region around digestive organs, and gonadal WAT (GWAT; male epididymal WAT or female parametrial and periovarian WAT) which expands during the early stages of obesity [14] and serve as a vital adipose depot for triglyceride storage [15]. WAT location is significant as increased visceral adipose mass is a risk factor related to metabolic disorders such as insulin resistance, while subcutaneous adipose tissue can be protective and improve insulin sensitivity [16,17,18]. BAT has been detected in the inter-scapular, subscapular, cervical, mediastinal, perirenal, pericardial, and periaortic regions in rodents [19] and humans [20,21,22,23,24]. In rodents, the commonly studied BAT is the interscapular BAT.

### 2.2. Cellular Properties of Adipose Tssues

WAT and BAT have very distinct histological and cellular properties. Conventional white adipocytes typically contain single large unilocular lipid droplets and few mitochondria. Conventional brown adipocytes typically consist of numerous multilocular lipid droplets and large amounts of mitochondria. Mitochondria contain high concentrations of iron pigmented-cytochromes, which gives a brownish color to BAT. Mitochondria of BAT feature uncoupling protein 1 (UCP1) in the inner membrane, a protein that uncouples mitochondrial respiration from ATP synthesis leading to energy released as heat [4,25], especially in the setting of a cold environment [3]. Thus, UCP1 is a unique biomarker for thermogenic BAT and BeAT.

Due to heterogeneity and plasticity of adipocytes, white and brown adipocytes change their morphology and function when energy demands are changed under certain physiological or pharmacological conditions. Specifically, in the setting of increased energy demand as seen in cold exposure, brown-like adipocytes develop and emerge in WAT, a process known as “browning”; whereas in a setting of decreased energy demands as seen in high-fat diet feeding, brown adipocytes can be converted to white-like adipocytes, a process termed as “whitening” [26,27,28]. The development of BeAT in WAT is considered favorable in systemic metabolic regulation [29,30]. A pioneering study reporting beige adipocytes in traditionally WAT first appeared in the literature 25 years ago [31]. Specifically, Young et al. reported enlarged BAT at classical BAT depots and presence of adipocytes with similar morphology and UCP1 content as brown adipocytes at parametrial WAT in cold-acclimated mice [31]. Loncar et al. then demonstrated increased mitochondria volume, crista density, and UCP1 expression in IWAT of cold-exposed cats [32] and mice [33], and such changes were reversible following exposure to a warm environment [33]. Cousin et al. further measured UCP1 mRNA and protein levels in several WAT depots in rats, and found that periovarian adipocytes displayed characteristics of brown adipocytes besides UCP1 content, such as increased density of mitochondrial crista [34]. A few research groups “rediscovered” thermogenic beige adipocytes in the recent decade [30,35,36,37]. Besides cold exposure, increased adrenergic signaling from treatment of adrenergic agonists also induces browning. Chronic administration of a selective β3-adrenergic receptor (β3-AR) agonist CL-316 243 triggers multilocular cells emerging in different WAT depots, including GWAT, RWAT, and IWAT, that normally contain only unilocular white adipocytes [38].

Beige, including brown-like and white-like, adipocytes exhibit histological and functional properties that are intermediary between white and brown adipocytes, containing numerous small lipid vacuoles with multilocular lipid droplets, surrounded by well-developed mitochondria [39,40,41,42,43,44,45]. One major difference is that, white-like beige adipocytes in BAT developed from whitening process, but not brown-like beige adipocytes in WAT developed from browning process, exhibit signs of inflammation such as increased crown-like structure formation and degenerating mitochondria [28]. As such, BeAT typically indicates the WAT that shelters brown-like beige adipocytes via browning process. Although there is a consensus that the presence of UCP1 expression in WAT is sufficient to define beige adipocytes and development of BeAT, it is unclear what other signatures can differentiate BeAT from typical BAT and how many beige cells must be present in a WAT depot to make it a BeAT.

Differential gene expression analysis of RNA sequencing data indicates that BAT expresses significantly greater level of *Ucp1* than GWAT (Padj < 0.05), but not RWAT or IWAT (Padj > 0.05), in lean and obese mice. Furthermore, four-week high-fat diet feeding does not significantly change *Ucp1* expression in BAT or WAT (Padj > 0.05; Table 1). These data confirm abundant expression of *Ucp1* in BAT and differing levels in WAT based on location. GWAT, which predominantly houses white adipocytes has the lowest level of *Ucp1* expression; while RWAT and IWAT, which are known to induce browning under certain conditions [46,47] contain mixed white and thermogenic adipocytes and thus have greater levels of *Ucp1* expression than GWAT (Padj < 0.05).

### 2.3. Precursors of Adipocytes

All types of adipocytes originate from multipotent mesenchymal stem cells. Various transcription factors induce differentiation of these stem cells into various types of precursor cells [48,49,50,51,52], ultimately driving cells to develop into adipocytes, myocytes, osteoblasts, chondrocytes, etc. Brown, but not white, adipocytes are derived from myogenic factor expressing progenitor cells of the central dermomyotome [48,49,50], and brown adipocytes share similar gene expression with myocytes. We have reported that BAT expresses genes involved in muscle development, structure, and contraction process such as those mesodermal developmental genes encoding myogenic factor 6 (*Myf6*), tropomyosin β (*Tpm2*), and sarcoglycan γ (*Sgcg*), and these genes are downregulated by high-fat diet feeding [12].

When expression levels of these genes are compared between WAT and BAT, BAT in lean mice expresses significantly higher levels of *Myf6*, *Tpm2*, and *Sgcg* than GWAT, RWAT, and IWAT (Padj < 0.05). In obese mice, the differences in expression of these genes by adipose tissue type are much less evident, with only *Tpm2* expression remaining significantly higher in BAT than GWAT and IWAT (Padj < 0.05), whereas expression of *Tpm2* between BAT and RWAT and expression of *Myf6* and *Sgcg* between BAT and WAT are similar (Padj > 0.05). The change of differential expression pattern comparing BAT versus WAT between lean and obese mice is due to high-fat diet significantly suppressing expression of *Myf6*, *Tpm2*, and *Sgcg* in BAT (Padj < 0.05) without changing expression of these genes in WAT (Table 1).

These findings confirm that brown adipocytes and myocytes share some common precursors [49] contributing to enriched expression of genes related to muscle differentiation and muscle function in BAT [53]. Additionally, depending on their locations, different types of adipocytes arise from different precursor cells and may have unique gene signatures. For example, white adipocytes found in visceral WAT versus subcutaneous WAT express different genes [17]. BAT has been shown to be composed of both brown and beige adipocytes [54]. In humans, brown adipocytes located at BAT depots relatively close to the body surface show gene signature more similar to beige adipocytes [41,55,56], while brown adipocytes found in BAT with deeper locations express classical brown adipocyte-selective markers [57]. Beige adipocytes display some unique molecular signatures that are not shared by either typical brown or white adipocytes. Furthermore, diminished and abolished differences in expression of myocyte-related genes between BAT and WAT by high-fat diet feeding in obese mice are possibly due to whitening process of brown adipocytes. Because white-like and brown-like adipocytes emerge within BAT and WAT, they may differentiate from brown or white adipocyte precursors or transdifferentiate from mature adipocytes.

### 2.4. Physiologic Functions of Adipose Tissues

The major function of WAT is energy storage and release. Energy is accumulated in the form of energy-rich lipid during times of positive energy balance, such as high-fat diet feeding. Stored energy is mobilized to release fatty acids and glycerol via lipolysis during times of negative energy balance, such as during exercise or fasting [3,4,5,6]. Thus, WAT plays significant roles in energy homeostasis via storing and releasing energy.

The key roles played by BAT in energy balance regulation has been known for about 40 years [58]. Although amount of active BAT in humans declines with increasing age, the presence of metabolically active BAT in adult humans was demonstrated over 20 years ago in clinical studies using 18F fluorodeoxyglucose and positron emission tomography/computed tomography imaging technology for the detection of cancerous tumors [59,60]. During the past decade, imaging studies not only have confirmed the presence of functional, metabolically active BAT that contributes to cold-induced thermogenesis in adult healthy humans, but also have revealed that BAT in adults can be activated physiologically or pharmacologically [20,21,22,23,61].

BAT and BeAT are thermogenic adipose tissues, which produce heat through energy dissipation for adaptive non-shivering thermogenesis in response to variety of stimuli [21,41,42,43,44,45,56,62,63,64]. A cold environment increases thermogenic capacity physiologically [21] and upregulates genes involved in lipid metabolism in human BAT, suggesting that BAT activation by cold temperature enhances lipid metabolism in BAT [65]. Studies using genetic mouse models have consistently shown that increased amount and activity of BAT and BeAT would protect from body fat gain, and prevent or correct metabolic dysregulation induced by feeding a high-fat diet [66,67]. In contrast, dysfunction and reduced activity of BAT and BeAT decrease lipid metabolism and lead to obesity [21]. Thus, BAT and BeAT play essential roles in maintaining body temperature and regulating energy metabolism in small mammals, hibernating mammals, and humans [3,4,20,21,23]. BAT and BeAT are recognized as potential targets for increasing energy expenditure in the treatment or prevention of obesity, leading to increased research attention on BAT and BeAT due to its therapeutic potential in humans.

### 2.5. Secretory Factors of Adipose Tissues

WAT is not only a site for energy accumulation, but it also functions as an endocrine organ [4,5]. Since the seminal discovery of leptin [68], a number of adipose hormones secreted from WAT, known as adipokines, have been identified, and secretory functions of WAT has been confirmed [69,70]. When comparing expression of two well-defined WAT adipokines, leptin (*Lep*) [71] and adiponectin (*Adipoq*) [72], in WAT and BAT, differential gene expression analysis reveals similar patterns for *Lep* and *Adipoq*. First, expression of *Lep* or *Adipoq* is not significantly different between WAT and BAT in lean mice (Padj > 0.05). In contrast, each type of WAT expresses significantly greater levels of *Lep* and *Adipoq* than BAT of obese mice (Padj < 0.05). Second, high-fat diet-induced changes in expression pattern between WAT and BAT is due to enhanced *Lep* expression in GWAT, RWAT, and IWAT (Padj < 0.05); while a combination of slight suppression of *Adipoq* expression in BAT (Padj > 0.05) and no change at WAT (Table 1), consistent with increased leptin but reduced adiponectin circulating in obese mice [73]. These RNA sequencing data confirm that expression of these two well-defined WAT adipokines *Lep* and *Adipoq* changes with increases in adiposity [73] and are more abundant in WAT than BAT, especially in obese mice.

BAT releases batokines that are distinct from WAT adipokines and functions to facilitate metabolic processes that favor thermogenesis and induction of browning [74]. Some of identified batokines include fibroblast growth factor 21 (FGF21) [75] and neuregulin 4 (NRG4) [76]. These batokines is usually not elevated under basal condition, but are induced during brown adipogenesis, thermogenesis, or browning process. Interleukin 6 (IL-6) is a recently defined batokine from a BAT transplantation study. A few studies have shown improved glucose and lipid metabolism in mice with BAT transplantation [77,78,79,80] via enhancing sympathetic activity [77], a phenomenon that requires IL-6 release from transplanted BAT graft [80].

Although IL-6 has been proposed as a batokine [80], hard evidence is lacking. We compare the expression of IL-6 gene (*Il6*), along with genes encoding two well-identified batokines FGF21 (*Fgf21*) and NRG4 (*Nrg4*), between WAT and BAT. Expressions of batokine genes are usually upregulated during thermogenesis or browning, but not at basal conditions without cold exposure or stimulation of adrenal signaling. Like *Fgf21* and *Nrg4*, expression of *Il6* in BAT is similar between lean and obese mice. It is noteworthy that expression of *Nrg4* in GWAT, RWAT, and IWAT is significantly downregulated in high-fat diet-induced obesity (Padj < 0.05), consistent with a recent study [81]. Also similar to *Fgf21* and *Nrg4*, expression of *Il6* is abundant in BAT, but is not significantly different from WAT, in lean mice (Padj > 0.05). Different from *Fgf21* and *Nrg4*, expression of *Il6* is significantly higher in BAT than RWAT and IWAT in obese mice (Padj < 0.05; Table 1). These RNA sequencing data support the idea that IL-6 is a batokine dominantly expressed in BAT.

## 3. Innervation of Adipose Tissues

### 3.1. Innervation of Adipose Tissues Regulates Metabolism

The autonomic nervous system innervating metabolic tissues and organs consists of two major branches, the sympathetic and parasympathetic nervous system (SNS and PSNS, respectively), both of which are primary efferent pathways that involuntarily respond to endogenous and exogenous stimuli [82]. The SNS and its major neurotransmitter norepinephrine upregulate energy mobilization and usage, whereas PSNS and its major neurotransmitter acetylcholine upregulate energy accumulation and storage. Obesity is associated with reduced sympathetic activity or lowered sympathetic response [83]. Sympathetic activity in WAT and BAT in response to different stimuli, such as administration of glucose or insulin [84], cold [21], and physical exercise [85] becomes blunted in obese individuals, which further promotes weight gain [86]. Characterization of neural circuits from the brain to adipose tissues has represented a challenging issue. Below we provide more details regarding the innervation of adipose tissues.

### 3.2. Innervation of Adipose Tissues from a Historic View

BAT innervation was reported many years before the innervation of WAT, with the first report of innervation of BAT at pericardial region appearing in late 1890s [87]. BAT innervation in rodents was then demonstrated by multiple research groups between 1930s and 1960s with differing opinions on the function of innervation, due to the fact that nerve fibers of BAT were seen both at the parenchymal space assumedly innervating brown adipocytes [88] and at vasculature assumedly innervating blood vessels [89]. BAT innervation was further studied using electron microscopy, and both myelinated and unmyelinated nerve fibers were seen at the parenchymal space and around vessels of BAT [90]. Importantly, axon terminals of some unmyelinated nerves containing synaptic vesicles were embedded on the surface of adipocytes [90]. This evidence showed brown adipocytes in direct contact with parenchymal nerves, and thus direct innervation of brown adipocytes has been well accepted ever since.

The innervation of WAT has been relatively understudied due to high lipid content in WAT that makes innervation difficult to visualize. It is also challenging to distinguish if terminals synapse on white adipocytes or stromal vascular fractions including vascular endothelial cells of vasculature, fibroblasts, immune cells such as macrophages. Similar to BAT innervation, debate for WAT innervation has focused on the nerve fibers around vasculature. Earlier studies suggested that WAT nerve fibers mostly were perivascular, and white adipocytes were not directly innervated or receive sparse neural inputs [91,92]. It remained uncertain if WAT innervation regulates vascular function or lipid metabolism until the 1990s. Since then much progress has been made in the understanding of different types of innervation of adipose tissues along with its importance in the regulation of energy metabolism.

### 3.3. Sympathetic Innervation

The synaptic vesicles at nerve axon terminals were presumed to contain catecholamines and be of sympathetic origin in early studies [90]. Influential studies by Bartness and colleagues have provided abundant anatomical, histological, biochemical, and functional evidence that have advanced the field of adipose tissue innervation [93]. In early 1990s, Youngstrom and Bartness injected a retrograde fluorescent tract tracer FluoroGold into IWAT and GWAT, and FluoroGold-labeled cells were observed in sympathetic ganglia T13 and T13-L2, respectively. They also injected an anterograde fluorescent tract tracer indocarbocyanine perchlorate into sympathetic ganglion T13, and many labeled cells were seen in the extracellular space surrounding adipocytes in IWAT and GWAT [92]. This pivotal study provided undisputable proof of sympathetic innervation of white adipocytes and not just their vasculature [92].

Bartness and colleagues then demonstrated brain sympathetic neuronal connectivity with WAT and BAT [94,95] using a retrograde tracer, pseudorabies virus (PRV), which only traces sympathetic neurons that are synaptically connected [96]. This method defines hierarchical connectome mapping neuronal pathways from the brain to adipose tissues. They further demonstrated that many PRV-labeled neurons in important regions of the hypothalamus and brainstem involved in regulation of feeding and energy expenditure express tyrosine hydroxylase (TH), a rate-limiting enzyme, and dopamine β-hydroxylase, the final enzyme, for biosynthesis of norepinephrine. This not only indicated that the labeled circuit marks the sympathetic innervation, but also supported a role for sympathetic innervation of adipose tissues in energy metabolism [97]. Recent studies have demonstrated that different locations of WAT and BAT are governed by diverse neurons and receive different degrees of sympathetic innervation from the brain regions involved in metabolic regulations, with more sympathetic neurons innervating BAT than IWAT throughout the entire neuroaxis [98].

Besides the tracing studies, sympathetic innervation of adipose tissues has been demonstrated using immunohistochemical labeling for sympathetic nerves marker TH using thin adipose sections of <10 µM thickness [77,91,99,100,101]. Immunohistochemical staining for TH in subcutaneous WAT, visceral WAT, and BAT has shown that TH immunoreactivity at parenchymal nerve fibers increases following cold exposure, which is accompanied by increased UCP1 expression and sympathetic activity [46,77,101,102,103,104]. Warm temperature elicits the opposite effect in BAT, with decreased sympathetic nerve activity and TH immunoreactivity [105].

### 3.4. Sensory Innervation

Anterograde tracers have been used to trace sensory nerve projections from adipose tissues to dorsal root ganglia that house cell bodies of sensory neurons. Following implantation of True Blue, an anterograde neural tracer, into subcutaneous IWAT, it appears in T13-L3 dorsal root ganglia, indicating sensory innervation of IWAT [106]. H129 strain of herpes simplex virus, an anterograde transneuronal tract tracer, has also been used to trace sensory nerve projections from IWAT and GWAT through T13-L1 dorsal root ganglia [107] and from interscapular BAT through C1-T4 dorsal root ganglia [108] in Siberian hamsters. Besides the tracing studies, sensory innervation of adipose tissue has been confirmed by histologically marking sensory nerves with sensory-associated neuropeptides, such as calcitonin gene-related peptide (CGRP) and substance P [91,99,100,109].

WAT sensory innervation is understudied compared to its sympathetic innervation, and its function is not completely understood. It has been proposed that sensory innervation may convey information about adiposity to the brain as an afferent pathway and communicate between the brain and adipose tissues as a feedback mechanism to regulate efferent sympathetic output [107,108,110]. Additionally, it is unclear what stimulates the secretion of CGRP or substance P in WAT or BAT and how these sensory nerve-associated neuropeptides affect energy balance. Nevertheless, the presence of both sympathetic and sensory nerves in adipose tissues supports a two-way communication with the brain, through afferent sensory and efferent sympathetic fibers (Figure 1).

### 3.5. Parasympathetic Innervation

All BAT depots receive sympathetic innervation, but only mediastinal and pericardial BAT has parasympathetic innervation [26,111]. Currently it is generally accepted that WAT has negligible or no parasympathetic innervation, but there was a debate around this topic in early 2000s.

Kreier et al. used PRV to mark parasympathetic nerves in rats and suggested the presence of parasympathetic nerves in WAT [112], which was questioned by Giordano et al. when WAT histology failed to label parasympathetic postganglionic nerve markers such as vesicular acetylcholine transporter (VAChT), vasoactive intestinal protein, and neuronal nitric oxide synthase in Siberian hamsters [113]. The marked CNS parasympathetic neurons and vagal innervation could be false positives due to leaking of viral tracer PRV or improper surgical procedure [114]. Kreier et al. responded that lack of parasympathetic nerve marker staining was inadequate to rule out parasympathetic innervation, as these markers were not stable [115]. Berthoud et al. [116] and Giordano et al. [117] further questioned the presence of parasympathetic innervation due to absence of any parasympathetic innervation-related markers for ganglia, nerve, neurotransmitters, etc. at various types of WAT from multiple species tested including mice, rats, and Siberian hamsters.

Recent advances in whole-tissue three-dimensional imaging of adipose nerve fibers using multiphoton microscopy clearly demonstrate that axons project to white adipocytes and form neuro-adipose synaptic connections in the parenchyma of adipose tissues [2,118,119,120], which helps to clear up confusion and increases our understanding of nerve-adipocyte and nerve-vasculature interactions. A seminal study labels three markers, an adipocyte marker perilipin, a neural pre-synaptic marker synaptophysin, and a sympathetic (TH) or a parasympathetic (VAChT) nerve marker in IWAT, and visualizes and quantifies different types of nerves using this whole-tissue three-dimensional imaging method [121]. This study shows that sympathetic nerve fibers (labeled with TH) are located in close contact with approximately 91.3% of all adipocytes (labeled with perilipin), and 98.8% of the neural fibers (labeled with synaptophysin) are labeled with TH, suggesting over 90% of adipocytes are being directly innervated and nearly 99% of that innervation being sympathetic in IWAT. Additionally, this study helps to clarify the issue of the presence of parasympathetic innervation in WAT, an unsettled controversial issue over the previous decade. Less than five nerve fibers labeled with parasympathetic nerve marker VAChT are detected in each IWAT sample, in contrast to the extensive arborization of fibers labeled with TH [121], supporting the idea of pervasive sympathetic innervation with little to no parasympathetic fibers in WAT [113].

Furthermore, because of the arborization pattern of sympathetic innervation, imaging of whole-tissue, but not thin-sections, is necessary for accurate quantification of innervation. It is worth noting that histology using adipose thin sections is suitable to visualize gross morphology and cellularity such as cell size and cell number related to adiposity change such as hypertrophy versus hyperplasia, cell structure change such as multilocularity versus unilocularity during browning or whitening, expression of UCP1 and nerve markers, and presence of macrophages in crown-like structures; but not suitable to quantify innervation, as cross-sections of nerves appear mostly as puncta, or visualization of synapses. Therefore, TH immunoreactivity in adipose tissues confirms the presence of sympathetic innervation, but does not assess quantity or activity of innervation, as expression of TH fluctuates in response to stimuli that change SNS activity.

### 3.6. Expression of Genes Related to Nerve Markers

RNA sequencing data from different types of adipose tissues of lean and obese mice allows for the analysis of genes encoding TH (*Th*), a sympathetic nerve marker; CGRP (*Calca*), a sensory nerve marker; and VAChT, a.k.a. solute carrier family 18 member 3 (*Slc18a3*), a parasympathetic nerve marker.

Differential gene expression analysis indicates that, expression of *Th* is generally abundant in BAT but not significantly different from WAT (Padj > 0.05). Expression of *Th* in BAT and GWAT is not significantly changed by high-fat diet feeding, but it is no longer detectable at RWAT and IWAT of obese mice, suggesting dramatic reduction in *Th* expression and thus sympathetic innervation in RWAT and IWAT during high-fat diet-induced obesity development. Expression of *Calca* is fairly abundant in WAT but not significantly different from BAT (Padj > 0.05). High-fat diet feeding does not significantly change *Calca* expression in any of WAT or BAT (Padj > 0.05). Interestingly, GWAT, RWAT, and IWAT have a tendency to express greater levels of *Calca* than BAT in obese mice (log2fold change > 1.5; Padj > 0.05), indicating that sensory innervation is relatively enhanced at WAT during obesity development, possibly to convey changes in lipid storage to the brain [110]. Bartness and colleagues have begun to understand the functional roles of adipose sensory innervation. They have reported that increased secretion of WAT adipokine leptin [122] and local WAT lipolysis [123] activate WAT sensory innervation, providing mechanisms for selective, depot-specific activation of WAT sensory innervation. These studies, supported by the RNA sequencing findings, indicate reduced sympathetic signal favors WAT lipid accumulation, with increased WAT sensory signal informing the brain of adiposity during obesity development, thus forming a two-way brain-adipose control of energy balance (Figure 1).

Expression of *Slc18a3* gene for solute carrier family 18 member 3 that transports acetylcholine is not detected in any of interscapular BAT, GWAT, RWAT, or IWAT (Table 1). It is noteworthy that both *Slc18a1* gene for solute carrier family 18 member 1 that transports serotonin and *Slc18a2* gene for solute carrier family 18 member 2 that transports norepinephrine, dopamine, serotonin, and histamine are expressed in all sequenced adipose tissue samples. The RNA sequencing data are consistent with a previous study showing that only mediastinal and pericardial BAT, but not interscapular BAT sampled in our study, have parasympathetic innervation [26]. The RNA sequencing data are also in accordance with imaging and histologic findings suggesting that parasympathetic innervation of WAT is negligible compared with sympathetic and sensory innervation [121].

## 4. Physiological Function of Adipose Tissues Sympathetic Innervation

### 4.1. Function of Sympathetic Innervation of WAT

Various stimuli that change energy fluxes modulate the SNS, which further regulates lipolysis, lipogenesis, adipocyte proliferation, and adipokine secretion [69,70,124]. Upon sympathetic stimulation, increases in firing rates of the sympathetic neurons that innervate adipose tissues induce secretion of norepinephrine at sympathetic postganglionic nerve terminals of surrounding adipocytes, subsequently activating β3-AR, a G protein-coupled receptor [125] and adenylate cyclase, which increases intracellular levels of cyclic adenosine monophosphate (cAMP). cAMP functions as a second messenger and activates protein kinase A (PKA) [126].

In white adipocytes, PKA phosphorylates perilipin, a lipid droplet-associated protein, and a series of lipases and esterases that convert stored triglycerides to diacylglycerol and monoacylglycerol [126], and via a cascade of steps of lipolysis eventually to fatty acids and glycerol that can be used as energy fuels of other tissues [2] (Figure 3). Function of sympathetic innervation in lipolysis at WAT has been directly assessed using optogenetic nerve stimulation. An optical fiber is implanted unilaterally into subcutaneous IWAT, and nerve stimulation leads to norepinephrine release, phosphorylation of hormone-sensitive lipase (HSL), and increased fatty acid release upon stimulation, compared to contralateral un-stimulated IWAT [2]. This study provides direct functional evidence that activation of sympathetic fibers is adequate to facilitate norepinephrine release and lipolysis [2]. Furthermore, visceral WAT depots mobilize their lipid to a greater extent than subcutaneous WAT depots following sympathetic action. For example, lipolysis at RWAT shows great response to pharmacological stimulation by β3-AR agonists [127]. WAT sympathetic activation also inhibits adipocyte proliferation and WAT expansion, with greater inhibition of expansion in subcutaneous IWAT compared to visceral RWAT [128].

### 4.2. Function of Sympathetic Innervation of BAT

As mentioned previously, it has been known for over a century that there is sympathetic innervation of BAT [87]. Since then, studies have supported the idea that catecholamines drive BAT non-shivering thermogenesis that plays an integral role in body temperature maintenance in various mammalian species including humans [108,129,130,131,132,133,134]. Sympathetic activation of BAT thermogenesis promotes heat dissipation of the energy contained in triglycerides. In brown adipocytes, free fatty acids are transferred into mitochondria following sympathetic stimulation by carnitine palmitoyltransferase 1 (CPT1) located at the outer membrane of mitochondria. Inside the mitochondria, free fatty acids serve as fuel for β-oxidation, which produces NADH and FADH that are later oxidized in the electron transport chain. NE and β3-AR interaction and subsequent PKA-dependent processes lead to increased expression of UCP1, which disassociates the activity of the respiratory chain from ATP production, mediates proton reentry into mitochondrial membrane, converts energy of proton gradient into heat, and ultimately dissipates energy as heat [3,4,132,135] (Figure 3).

The role of sympathetic activity in browning of white adipocytes and the development of BeAT has been more recently described [30]. Various transcriptional regulators drive distinct steps of this process, leading to the differentiation of preadipocytes into brown or beige adipocytes, increased mitochondrial biogenesis, and overexpression of thermogenic proteins such as UCP1 [136]. For example, members of peroxisome proliferator-activated receptor (PPAR) family, CCAAT/enhancer-binding protein (C/EBP) family, and bone morphogenic protein family [48] are important transcriptional factors specific to adipocyte differentiation [56,137]. Specifically, activation of C/EBPβ, which cooperates with a dominant transcriptional co-regulator PR domain containing 16 (PRDM16) [138], induces PPARγ and C/EBPα expressed in preadipocytes, acts jointly to promote adipocyte differentiation and adipogenesis [137], which determines brown adipocyte lineage or enhances white adipocyte browning [139]. Members of PPAR family and C/EBP family have been implicated as key enhancers for browning of adipocytes and adipogenesis [27,140,141,142]. In addition, the hypothalamic AMP-activated protein kinase activates WAT sympathetic innervation to promote browning [104]. It is possible that members of PPAR family and C/EBP family at adipose tissues are regulated by sympathetic activity from certain stimuli, which subsequently controls adipocyte differentiation and proliferation as well as browning of white adipocytes. The events underlying browning processes regulated by distinct endogenous factors and environmental stimuli are far from fully established [136].

### 4.3. Comparison of Sympathetic Function and Gene Markers between WAT and BAT

Sympathetic innervation is differentially activated and performs differing functions in WAT and BAT. For example, in response to calorie overload, sympathetic activity in BAT increases to promote energy expenditure, whereas sympathetic activity in WAT decreases to promote lipogenesis and lipid accumulation. In contrast, in response to fasting or high-fat to low-fat diet switch, sympathetic activity in BAT, along with many other tissues, decreases to conserve energy expenditure, whereas sympathetic activity in WAT increases to promote lipid mobilization and fatty acid release for other tissues to use [103,143,144]. In response to a cold environment, sympathetic activity in both BAT and WAT increases to provide substrates for non-shivering thermogenesis to maintain body temperature [145,146]. Therefore, sympathetic regulation of lipid metabolism varies in WAT and BAT through discrete sympathetic projections [147], which diverge to coordinate WAT lipolysis and BAT thermogenesis, leading to finely tuned control of whole-body energy homeostasis [148].

Intracellularly, early steps upon sympathetic stimulation involving activation of β3-AR and adenylate cyclase, cAMP production, and PKA activation are common in both white and brown adipocytes. PKA then phosphorylates and activates specific intracellular target proteins in different type of cells. Lipolysis and fatty acid release involving activation of HSL is dominant in white adipocytes, and fatty acid oxidation involving CPT1 is dominant in brown adipocytes. Data from the comparison of the expression of HSL gene (*Lipe*) and CPT1 gene (*Cpt1b*) in WAT and BAT support lipolytic function of WAT and fatty acid oxidation of BAT as major mechanisms to reduce lipid accumulation and regulate whole-body lipid metabolism.

Differential gene expression analysis indicates that, high-fat diet feeding does not significantly change *Lipe* expression in WAT or BAT (Padj > 0.05). Expression of *Lipe* is similar between WAT and BAT in lean mice (Padj > 0.05), whereas subcutaneous IWAT, but not visceral GWAT or RWAT, expresses significantly greater expression of *Lipe* than BAT in obese mice (Padj < 0.05). These RNA sequencing data indicate more abundant *Lipe* expression in subcutaneous IWAT relative to BAT in high-fat diet fed obese mice, suggesting selective lipid mobilization at subcutaneous WAT but not at visceral WAT during calorie overload. This finding corroborates that lipid accumulation mostly occurs at visceral WAT during obesity development.

Differential gene expression analysis indicates that BAT expresses significantly greater *Cpt1b* than GWAT and RWAT in both lean and obese mice (Padj < 0.05), while BAT has a trend to express greater level of *Cpt1b* than IWAT (log2fold change > 1.5; Padj > 0.05). Additionally, there is a trend of upregulation of *Cpt1b* expression in BAT in obese mice (log2fold change > 1.5; Padj > 0.05) (Table 1). These data indicate more abundant *Cpt1b* expression in BAT than WAT, and support increased fatty acid oxidation at BAT leading to enhanced energy expenditure, contributing to the homeostatic regulation of whole-body adiposity during obesity development.

### 4.4. Studying Sympathetic Function of Adipose Tissues via Denervation

Denervation of WAT of BAT is a great tool for studying sympathetic functions, a topic that was comprehensively reviewed recently [149]. Denervation studies demonstrate metabolic perturbations following loss of the nerve supply to WAT or BAT. Surgical denervation of WAT increases adipose mass, reduces lipolysis, and promotes white adipocyte proliferation and differentiation in rats [150] and Siberian hamsters [92,99,100,128]. Surgical denervation of BAT decreases expression of TH [151] and UCP1 [152], and leads to impaired thermogenesis, reduced energy expenditure, increased body fat mass [153], and “whitening” of BAT [151]. These studies highlight the importance of WAT innervation in regulating lipolysis and proliferation, and BAT innervation in regulating thermogenesis and browning. Therefore, denervation studies add credence and further support neural control of metabolism by innervation of WAT and BAT.

Surgical denervation is considered more effective at eliminating neural input and output, as the nerve bundles are physically severed, while vasculature is left intact. Surgical denervation, however, is not specific to nerve type, as sympathetic and sensory nerves bundle and travel together, thus both are severed. Although surgical denervation can be used to differentiate neural and endocrine effects on metabolic regulation, it is not able to reveal which nerve type is relatively more essential in maintaining certain metabolic effects.

An alternative approach is chemical denervation that removes a selective type of nerve supply to WAT and BAT. Chemical sympathetic denervation typically uses 6-hydroxydopamine [154], which is taken up into norepinephrine storage vesicles, leading to oxidative damage to vesicle membrane and nerve degeneration, thereby producing denervation of sympathetic nerves while leaving sensory nerves intact [149]. Chemical sympathetic denervation reduces expression of TH and norepinephrine content without changing CGRP level, indicating intact sensory innervation, in BAT [98] and WAT [113]. Chemical sensory denervation typically uses 8-methyl-*N*-vanillyl-6-noneamide (i.e., capsaicin), the pungent principle of hot chili peppers, which over-activates vanilloid receptor (i.e., capsaicin receptor) and leads to influx of calcium and sodium ions, an excitotoxic effect that destroys small diameter nociceptive sensory neurons along with unmyelinated and myelinated sensory nerves [149]. Chemical sensory denervation reduces contents of CGRP and substance P, and leaves sympathetic nerves intact in BAT [155] and WAT [100,110]. Chemical denervation, however, has certain drawbacks including intra-tissue injuries related to multiple injections, uneven distribution of neurotoxins leading to lack of uniform damage, lower efficacy in eliminating TH- or CGRP-immunoreactivity compared to surgical denervation [100], and possible “reconnection” of damaged fibers as nerve markers reappear after a certain period of time following injection [156].

WAT sympathetic denervation results in increased adipose mass characterized by an increase in cell number and a decrease in lipolysis, which supports a mechanism by which the sympathetic nerves regulate lipid metabolism via regulating adipocyte proliferation and lipolysis [149]. Bilateral BAT sympathetic denervation increases sympathetic activity of IWAT [98], demonstrating adipose tissue crosstalk with the brain to maintain energy homeostasis. Similar to BAT chemical sympathetic denervation and in contrast to WAT sympathetic denervation, WAT sensory denervation increases adipose mass via hypertrophy instead of hyperplasia [99,100], providing a means of differential control of WAT by sympathetic versus sensory nerves. BAT sensory denervation also impairs thermogenesis [157], demonstrating the need for sensory feedback from BAT for proper thermogenic function. Therefore, chemical denervation helps to deepen the understanding of action of specific nerve types in adipose tissue, as well as feedback neurocircuit that involves the brain and multiple peripheral tissues and incorporates both afferent sensory and efferent sympathetic nerves.

## 5. Conclusions

Obesity manifested as excessive lipid storage in adipose tissue is due to metabolic dysregulation of energy homeostasis. WAT accumulates excess energy and BAT functions as a thermogenic organ in response to metabolic challenges such as diet and cold, thus both are vital for the regulation of lipid metabolism and body weight. The SNS plays a primary role in the regulation of BAT and BeAT thermogenic activation and fatty acid oxidation, as well as WAT lipolysis, lipogenesis, and browning (Figure 1). The functions of the SNS in these key metabolic processes represent potential therapeutic targets for treating obesity.

New whole-adipose imaging and genomic analysis have advanced understanding of adipose tissue innervation, transcriptome, and secretome. However, there is still much unknown regarding to gene signatures of different types of adipocytes; adipose tissue innervation, especially in terms of the innervation, transcriptome, and secretome of BeAT; how chemical messengers, including locally secreted WAT adipokines and BAT batokines, circulating hormones from other organs, neuropeptides and neurotransmitters from adjacent nerves, and affect sympathetic and sensory innervation of WAT and BAT; and feedback regulation to control lipid metabolism. Additionally, the concept of brain regulating lipid metabolism via innervation of WAT and BAT is well accepted, but most of current knowledge is obtained from animal studies using rodents. It is uncertain if humans have similar afferent and efferent neural circuits and feedback regulation of adipose tissues. More studies are needed to identify neuroanatomic and synaptic structures of, metabolic functions, and chemical messengers released from and used by each type of nerves in WAT and BAT. Now is an exciting time for further research in neuroendocrine regulation of lipid metabolism at different types of adipose tissues.

## Figures and Tables

**Figure 1 ijms-20-02707-f001:**
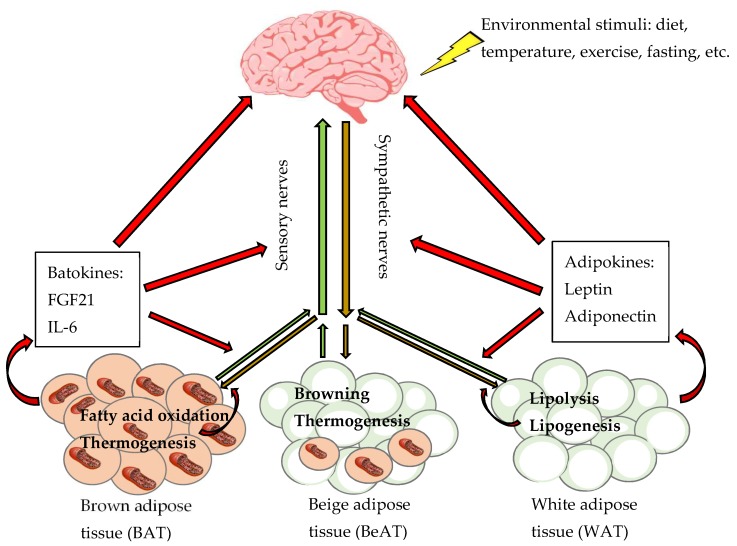
Neuroendocrine regulation of lipid metabolism at different types of adipose tissues. A two-way communication exists between the brain and white, brown and beige adipose tissues (WAT, BAT and BeAT, respectively) involving neural signals consisting of afferent sensory nerves (green arrows) and efferent sympathetic nerves (brown arrows), and endocrine signals (red arrows) consisting of WAT adipokines and BAT batokines. Environmental stimuli (i.e., diet, temperature, exercise, fasting) that change energy stores modulate sympathetic activity to regulate lipolysis, lipogenesis, and adipokine secretion at WAT; thermogenesis, fatty acid oxidation, and batokine secretion at BAT; and induction of BeAT and browning. A number of WAT adipokines (such as leptin and adiponectin) and BAT batokines (such as fibroblast growth factor 21 [FGF21] and interleukin 6 [IL-6]) interact with neural circuits to cooperatively regulate whole-body energy metabolism.

**Figure 2 ijms-20-02707-f002:**
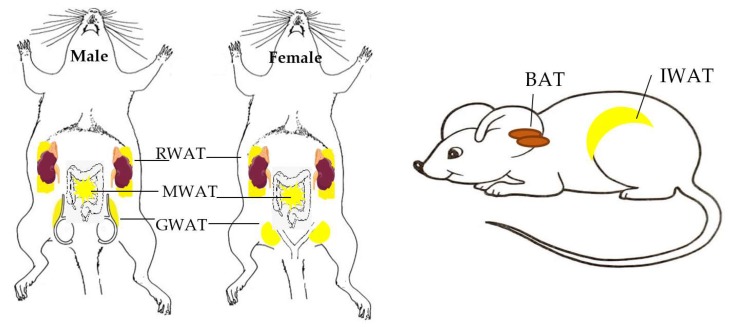
Schematic overview of location of different types of white and brown adipose tissues. Schematic diagrams indicating locations of different types of commonly studied WAT and BAT commonly studied in male and female rodent models, including visceral RWAT, MWAT, and GWAT, as well as subcutaneous IWAT and interscapular BAT.

**Figure 3 ijms-20-02707-f003:**
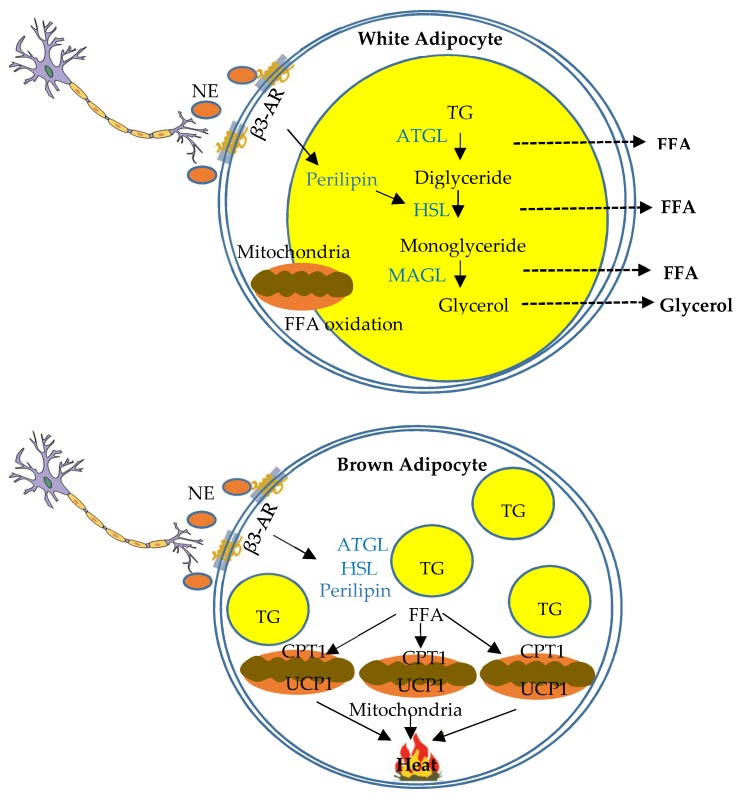
Schematic overview of function of sympathetic innervation of white and brown adipose tissues in the regulation of lipid metabolism. Schematic diagram indicating lipolysis and thermogenesis in white and brown adipocytes respectively regulated by sympathetic innervation. In white adipocytes, sympathetic innervation regulates a cascade of lipolysis to convert stored triglycerides (TG) to free fatty acids (FFA) and glycerol that can be used as fuels of other tissues. In brown adipocytes, sympathetic innervation regulates non-shivering thermogenesis. FFAs are transferred into mitochondria, primarily by carnitine palmitoyltransferase 1 (CPT1), and serve as fuel for β-oxidation. UCP1 is activated to disassociate respiratory chain from ATP production, and ultimately dissipates energy as heat.

**Table 1 ijms-20-02707-t001:** Genes compared between white adipose tissue (WAT) and brown adipose tissue (BAT) of lean and obese male mice using RNA sequencing data.

Gene Categories	Genes	WAT		BAT	
		Lean	Obese	Lean	Obese
thermogenesis	uncoupling protein 1 (*Ucp1*)	Low	Low	High	High
Brown adipocyte precursors	myogenic factor 6 (*Myf6*)	Low	NS	High	NS −
	sarcoglycan gamma (*Sgcg*)				
	tropomyosin β (*Tpm2*)	Low	Low	High	High −
WAT adipokines	leptin (*Lep*)	NS	High +	NS	low
	adiponectin (*Adipoq*)	NS	High	NS	low
BAT batokines	interleukin 6 (*Il6*)	Low	Low	High	High
	fibroblast growth factor 21 (*Fgf21*)	NS	NS	NS	NS
	neuregulin 4 (*Nrg4*)	NS	NS −	NS	NS
Sympathetic nerve	tyrosine hydroxylase (*Th*)	Low	NS/ND	High	NS
Sensory nerve	calcitonin gene-related peptide (*Calca*)	NS	High	NS	Low
Parasympathetic nerve	vesicular acetylcholine transporter (*Slc18a3*)	ND			
Lipolysis	hormone-sensitive lipase (*Lipe*)	NS	High	NS	Low
Fatty acid oxidation	carnitine palmitoyltransferase 1 (*Cpt1b*)	Low	Low	High	High +

Obese mice fed with a high-fat diet had three times of adiposity comparing to the lean mice fed with a low-fat diet for four weeks. Approximately 30 million of single-end sequencing reads were retrieved from each WAT and BAT sample and aligned to the mouse genome (ENSEMBL 84 release, GRCm38.p4) [12]. The sequence data are available at a publicly accessible database, Gene Expression Omnibus (GEO; https://www.ncbi.nlm.nih.gov/geo/info/seq.html). The accession numbers are GSE112740 for BAT samples and GSE112999 for WAT samples. The differentially expressed genes between lean and obese mice were analyzed using DESeq2 package in R statistical language. Threshold for differential expressed genes is absolute value log2 fold change > 1.5 and Padj < 0.05 is considered statistically significant. Low: at least one type of WAT has significantly lower expression than BAT. High: at least one type of WAT has significantly higher expression than BAT. NS: not significantly different expression between BAT and any type of WAT. ND: Not detected. +: upregulated in obese mice. −: downregulated in obese mice.

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
