# Peer review of "Neuroendocrine Regulation of Energy Metabolism Involving Different Types of Adipose Tissues"

_ijms, 2019, doi:10.3390/ijms20112707_

Reviewer 1 Report

The manuscript IJMS-505772 describe the neuroendocrine regulation of adipose function and the role that the different types of adipose tissues play in maintaining energy homeostasis in response to the interaction between adipokines/batokines and neural signals.

A few minor point should be further emphasized by the authors:

-           As reported by the authors, PPARγ is considered an important regulator of adipogenesis, but    what is the function of C/EBP in the WAT and BAT development?

-           What is the role of adipose tissue metabolism for insulin resistance? And in aging?

Author Response

Reviewer 2:

The manuscript IJMS-505772 describe the neuroendocrine regulation of adipose function and the role that the different types of adipose tissues play in maintaining energy homeostasis in response to the interaction between adipokines/batokines and neural signals.

A few minor points should be further emphasized by the authors

-           As reported by the authors, PPARγ is considered an important regulator of adipogenesis, but what is the function of C/EBP in the WAT and BAT development?

We agree with the Reviewer and include the discussion of the function of C/EBP in adipocyte differentiation and adipogenesis on Page 12 of this revision.

The paragraph now reads as below:

The role of sympathetic activity in browning of white adipocytes and the development of BeAT has been more recently described (Harms and Seale 2013). Various transcriptional regulators drive distinct steps of this process, leading to the differentiation of preadipocytes into brown or beige adipocytes, increased mitochondrial biogenesis, and overexpression of thermogenic proteins such as UCP1 (Montanari, Pošćić et al. 2017). For example, members of peroxisome proliferator-activated receptor (PPAR) family, CCAAT/enhancer-binding protein (C/EBP) family, and bone morphogenic protein family (Tseng, Kokkotou et al. 2008) are important transcriptional factors specific to adipocyte differentiation (Gregoire, Smas et al. 1998, Wu, Cohen et al. 2013). Specifically, activation of C/EBPβ, which cooperates with a dominant transcriptional co-regulator PR domain containing 16 (PRDM16) (Kajimura, Seale et al. 2009), induces PPARγ and C/EBPα expressed in preadipocytes, acts jointly to promote adipocyte differentiation and adipogenesis (Gregoire, Smas et al. 1998), which determines brown adipocyte lineage or enhances white adipocyte browning (Fukunaka, Fukada et al. 2017). Members of PPAR family and C/EBP family have been implicated as key enhancers for browning of adipocytes and adipogenesis (Rosen, Sarraf et al. 1999, Karamanlidis, Karamitri et al. 2007, Fisher, Kleiner et al. 2012, Giralt and Villarroya 2013). In addition, the hypothalamic AMP-activated protein kinase activates WAT sympathetic innervation to promote browning (Vitali, Murano et al. 2012). It is possible that members of PPAR family and C/EBP family at adipose tissues could be regulated by sympathetic activity from certain stimuli, which subsequently controls adipocyte differentiation and proliferation as well as browning of white adipocytes. The events underlying browning processes regulated by distinct endogenous factors and environmental stimuli are far from fully established (Montanari, Pošćić et al. 2017).

-           What is the role of adipose tissue metabolism for insulin resistance? And in aging?

The role of adipose tissue metabolism in insulin resistance or aging is a broad topic and is beyond the focus of this review.  

Fisher, f. M., S. Kleiner, N. Douris, E. C. Fox, R. J. Mepani, F. Verdeguer, J. Wu, A. Kharitonenkov, J. S. Flier, E. Maratos-Flier and B. M. Spiegelman (2012). "FGF21 regulates PGC-1α and browning of white adipose tissues in adaptive thermogenesis." Genes Dev 26(3): 271-281.

Fukunaka, A., T. Fukada, J. Bhin, L. Suzuki, T. Tsuzuki, Y. Takamine, B.-H. Bin, T. Yoshihara, N. Ichinoseki-Sekine, H. Naito, T. Miyatsuka, S. Takamiya, T. Sasaki, T. Inagaki, T. Kitamura, S. Kajimura, H. Watada and Y. Fujitani (2017). "Zinc transporter ZIP13 suppresses beige adipocyte biogenesis and energy expenditure by regulating C/EBP-β expression." PLOS Genetics 13(8): e1006950.

Giralt, M. and F. Villarroya (2013). "White, brown, beige/brite: different adipose cells for different functions?" Endocrinology 154(9): 2992-3000.

Gregoire, F. M., C. M. Smas and H. S. Sul (1998). "Understanding adipocyte differentiation." Physiol Rev 78(3): 783-809.

Harms, M. and P. Seale (2013). "Brown and beige fat: development, function and therapeutic potential." Nat Med 19(10): 1252-1263.

Kajimura, S., P. Seale, K. Kubota, E. Lunsford, J. V. Frangioni, S. P. Gygi and B. M. Spiegelman (2009). "Initiation of myoblast to brown fat switch by a PRDM16–C/EBP-β transcriptional complex." Nature 460: 1154.

Karamanlidis, G., A. Karamitri, K. Docherty, D. G. Hazlerigg and M. A. Lomax (2007). "C/EBPbeta reprograms white 3T3-L1 preadipocytes to a brown adipocyte pattern of gene expression." J Biol Chem 282(34): 24660-24669.

Montanari, T., N. Pošćić and M. Colitti (2017). "Factors involved in white-to-brown adipose tissue conversion and in thermogenesis: a review." Obes Rev 18(5): 495-513.

Rosen, E. D., P. Sarraf, A. E. Troy, G. Bradwin, K. Moore, D. S. Milstone, B. M. Spiegelman and R. M. Mortensen (1999). "PPAR gamma is required for the differentiation of adipose tissue in vivo and in vitro." Mol Cell 4(4): 611-617.

Tseng, Y.-H., E. Kokkotou, T. J. Schulz, T. L. Huang, J. N. Winnay, C. M. Taniguchi, T. T. Tran, R. Suzuki, D. O. Espinoza, Y. Yamamoto, M. J. Ahrens, A. T. Dudley, A. W. Norris, R. N. Kulkarni and C. R. Kahn (2008). "New role of bone morphogenetic protein 7 in brown adipogenesis and energy expenditure." Nature 454(7207): 1000-1004.

Vitali, A., I. Murano, M. C. Zingaretti, A. Frontini, D. Ricquier and S. Cinti (2012). "The adipose organ of obesity-prone C57BL/6J mice is composed of mixed white and brown adipocytes." J Lipid Res 53(4): 619-629.

Wu, J., P. Cohen and B. M. Spiegelman (2013). "Adaptive thermogenesis in adipocytes: Is beige the new brown?" Genes Dev 27(3): 234-250.

Reviewer 2 Report

Authors have presented an elegant story where they describe potential role of various adipose tissue and their interplay against energy metabolism and neural signals. Authors have taken utmost care to provide a didactic story which in detail introduces and explains the origin, properties, functions and dynamics of various forms of adipose tissue, later making a strong case for innervation - are the merit of this manuscript. 

Addition of more pics or cartoons not only increase the aesthetic value for a review article but also increase the readership. Authors are required to provide a cartoon (human body) which would identify various locations of BAT, WAT and BeAT, as explained tine section 2.1

Similarly, authors are required to furnish another simplistic cartoon for the innervation part (section 3). Although the Fig 1 gives a brief introduction, it does not serve for section 3. A cartton for section 3 should be the most important pic/conclusion/message of this manuscript.

Author Response

We thank the reviewers for their valuable comments, which we used to improve the manuscript. We believe that changes of this revision increase the readership of this manuscript.

Reviewer 1:

Authors have presented an elegant story where they describe potential role of various adipose tissue and their interplay against energy metabolism and neural signals. Authors have taken utmost care to provide a didactic story which in detail introduces and explains the origin, properties, functions and dynamics of various forms of adipose tissue, later making a strong case for innervation - are the merit of this manuscript.

Addition of more pics or cartoons not only increase the aesthetic value for a review article but also increase the readership. Authors are required to provide a cartoon (human body) which would identify various locations of BAT, WAT and BeAT, as explained in section 2.1

We agree with the Reviewer and included a figure (Figure 2) on Page 3 of this revision showing locations of different types of adipose tissues. The literature discussed in the review are mostly from animal studies using rodent models. To avoid confusion, we include a schematic figure indicating locations of different adipose tissue depots in rodents

Similarly, authors are required to furnish another simplistic cartoon for the innervation part (section 3). Although the Fig 1 gives a brief introduction, it does not serve for section 3. A cartoon for section 3 should be the most important pic/conclusion/message of this manuscript.

We agree with the Reviewer that a figure (Figure 3) on Page 11 of this revision showing sympathetic function, regulation of lipolysis and thermogenesis within white and brown adipose tissues respectively. This figure would complement to the figure 1 that shows neural circuits that provide feedback regulation between the CNS and adipose tissues.